# Natural soundscapes enhance mood recovery amid anthropogenic noise pollution

Lia R. V. Gilmour[1], Isabelle Bray[2], Chris Alford[2,3], Paul R. Lintott[1] *

**1** School of Applied Sciences, College of Health, Science and Society, University of the West of England, Bristol, United Kingdom, **2** School of Health and Social Wellbeing, College of Health, Science and Society, University of the West of England, Bristol, United Kingdom, **3** Centre for Human Psychopharmacology, Swinburne University of Technology, Melbourne, VIC, Australia

* Paul.Lintott@uwe.ac.uk

## Abstract

In urbanised landscapes, the scarcity of green spaces and increased exposure to anthropogenic noise have adverse effects on health and wellbeing. While reduced speed limits have historically been implemented to address traffic safety, their potential impact on residents' wellbeing, especially in relation to engagement with natural soundscapes, remains understudied. Our study investigates the influence of i) natural soundscapes, including bird song, and ii) the addition of traffic noise to natural soundscapes at two speeds (20 mi/h and 40 mi/h) on mood. We found that natural soundscapes were strongly linked with the lowest levels of anxiety and stress, with an increase in stress levels associated with mixed natural soundscapes with the addition of 20 mi/h traffic noise and the highest levels with 40 mi/h traffic noise. Higher levels of hedonic tone, indicative of positive mood, was noted with natural soundscapes, but diminished when combined with 40 mi/h traffic noise. Our results show that anthropogenic soundscapes including traffic sounds can mask the positive impact of natural soundscapes including birdsong on stress and anxiety. However, reducing traffic speeds in cities could be a positive intervention for enhancing access to nature. Technological solutions, such as the widespread adoption of hybrid and electric vehicles, and urban planning strategies like integrating green spaces into transit routes, offer potential opportunities to mitigate the impact of noise pollution and benefit humans in urban environments.

## Introduction

Since the industrial revolution, many landscapes have become increasingly characterised by their anthropogenic footprint and this has profoundly affected both human health and wellbeing, as well as ecological communities [1]. Access to nature can lead to reduced risk of obesity and dementia, increase life satisfaction, aid stress recovery and provide restorative benefits (e.g. [2–5]. Furthermore, faced with global biodiversity and climate crises, increasing connection between people and nature is also important for improving engagement in environmental conservation actions [6]. Consequently, better understanding of the relationship between humans and nature in urban areas is vital.

**Competing interests:** The authors have declared that no competing interests exist.

Beneficial effects of nature may depend on an individual's background, perception of biodiversity, its restorative potential and the senses used to perceive it [5,7]. Recently, various soundscape playback studies have examined the potential impacts of exposure to natural sounds (reviewed by [8]) and have shown that natural soundscapes, can aid health recovery and attention restoration in participants [9–14]. Studies have reported both physiological and psychological benefits to health from listening to natural sounds or soundscapes [8]. For example, natural sound exposure has been shown to lower blood pressure, heart and respiratory rates, as well as self-reported stress and anxiety [8,13,15]. Listening to natural sounds can also improve attention restoration and increase cognitive performance [8,13,14]. Responses to nature, including natural soundscapes, may also be dependent on factors such as age, sex and socio-cultural experience e.g. childhood connection to nature [11,16,17], though more research is needed to understand and tease apart the influence of these factors.

Conversely, anthropogenic soundscapes, such as those including traffic or aircraft noise, can have negative effects on human health and wellbeing with physiological and psychological effects recorded across the literature [18–22]. Exposure to sounds made by traffic infrastructure (including rail, road and air traffic noise) has been recorded to increase the risk of depression [21], severe anxiety [22] and physiological stress responses such as increase in cortisol levels [18]. Indeed, psychological effects of listening to traffic are likely to be caused by direct adverse impacts and changes in the central nervous system, causing for example changes to brain tissue and neuroinflammation [23]. Long-term effects of exposure to traffic noise also include increased instances of cardiovascular disease, risk of stroke, diabetes, hypertension and loss of hearing [19]. Higher traffic speeds are associated with higher noise pollution and perceived annoyance levels in people exposed to the sounds [24]. However, to our knowledge there has been no study to date that has examined the impact of lowering road traffic speeds on the sonic environment people are exposed to in urban environments and how this affects wellbeing.

Historically, reduced speed limits have been introduced to slow traffic and reduce the number and severity of road traffic collisions. International agencies including the World Health Organisation, World Bank and the Organisation for Economic Co-operation and Development have promoted the implementation of reduced speed limits across residential roads (e.g. reducing speed limits in the UK from 30 mi/h down to 20 mi/h, equating to a reduction of 50 to 30 km/h in equivalent traffic regimes). Although reduced traffic speeds are strongly associated with reductions in road traffic casualties (e.g. [25]), the current move towards lower speed limits is only in part powered by the injury prevention agenda. Reduced speed limits lessen noise pollution, as well as having proportionately larger positive impacts for poorer communities which suffer from higher levels of traffic pollution [26], thereby reducing inequalities in health. Also significant is the desire to reduce obesity though the promotion of physical activity in the form of walking, cycling and active play [26]. A reduction in background noise may also allow urban populations to hear wildlife more clearly, however the potential benefits of reducing speed limits on wellbeing amongst residents has not routinely been researched.

In this study, we aimed to test whether listening to natural soundscapes can aid stress recovery and reduce anxiety in student participants exposed to noise pollution and whether reducing traffic speeds affects psychological responses to these natural sounds. Specifically, we aimed to use two widely applied subjective scales commonly used to measure human mood to test to what extent self-reported anxiety, stress and pleasure are affected by i) natural soundscapes and ii) mixed natural and anthropogenic soundscapes, with the addition of traffic noise recorded on roads with two common speed limits in the UK, 20 mi/h and 40 mi/h (e.g. similar to 30 and 60 km/h). We also tested whether general mood state, age, gender and inherent preference for natural environments affected mood recovery in response to natural and mixed natural and anthropogenic soundscapes.

## Methods

### Participants

Participants (n = 68) were recruited from the University of the West of England (UWE) Psychology participant pool, online on the UWE student's union survey page and via email between 5th March-14th April 2021. Written informed consent was obtained from all participants prior to their participation and they were given the option to withdrawn from the experiment at any time. Psychology students that participated received an incentive of one course credit, but other students were not incentivised. All students that took part were either science, psychology or social science students. We excluded participants that had been diagnosed with or currently taking prescribed medication for psychiatric conditions such as anxiety and depression, as low mood and medication could influence subjective responses. We also wanted to avoid inducing increased stress response in those already suffering from anxiety and depression. Ethical approval was obtained from the university ethics committee prior to the experiment commencing (University of the West of England approval code: HAS.20.11.036).

### Soundscapes

We created three 3-minute soundscape files for use in playback experiments using Audacity (v.2.4.2, Audacity Team, open-source software, https://www.audacityteam.org/). All three soundscapes included a base of a natural soundscape recording, made at sunrise in West Sussex, UK using a parabolic reflector microphone system (Talinga, Tobo, Sweden). The natural soundscape included a range of common UK bird species likely to be heard in a typical dawn chorus including those recorded on the RSPB's top 25 Big Garden Birdwatch [27] and some rarer species such as the nightingale (*Luscinia megarhynchos*). We added anthropogenic soundscapes including traffic noise to the natural soundscape to create the other two *mixed* natural and anthropogenic soundscapes. Anthropogenic soundscapes including road traffic sounds were recorded on two roads (20 mi/h and 40 mi/h limits) in Bath, UK on the same morning at peak 'rush hour' between 8.30–9.00 am, using a Zoom H5 portable recorder (Zoom Inc. Tokyo, Japan) at 1 meter from each road. Traffic volume was consistently high between the two roads and contained rush hour traffic of mainly cars and buses. Soundscapes used in playback experiments therefore included i) a natural soundscape ('natural'), ii) a mixed natural + anthropogenic soundscape with 20 mi/h traffic ('mixed 20') and iii) a mixed natural + anthropogenic soundscape with 40 mi/h traffic ('mixed 40').

### Experimental design

Data were collected using a bespoke survey designed and administered online using Qualtrics XM software (Qualtrics International, Seattle, United States). Participants were given a basic background and information about the experiment, including that it contained a task to listen to soundscapes and watch videos, but not given any information about the hypotheses (see S1 Doc for exemplar experiment procedure). Participants took part in the experiment in one sitting online for 30 minutes to 1 hour (depending on the time taken to answer questions) (Fig 1). Each participant was exposed to three rounds of a stressor video for 1 minute and soundscape play back of 3 minutes and answered questions after each stressor and each soundscape (12 minutes of exposure to stressors and soundscape in total). At the end of the playback experiment, participants were asked demographics questions (age, gender, ethnicity) and some other questions to gain an understanding of any participant bias (see *Demographics and other participant information*).

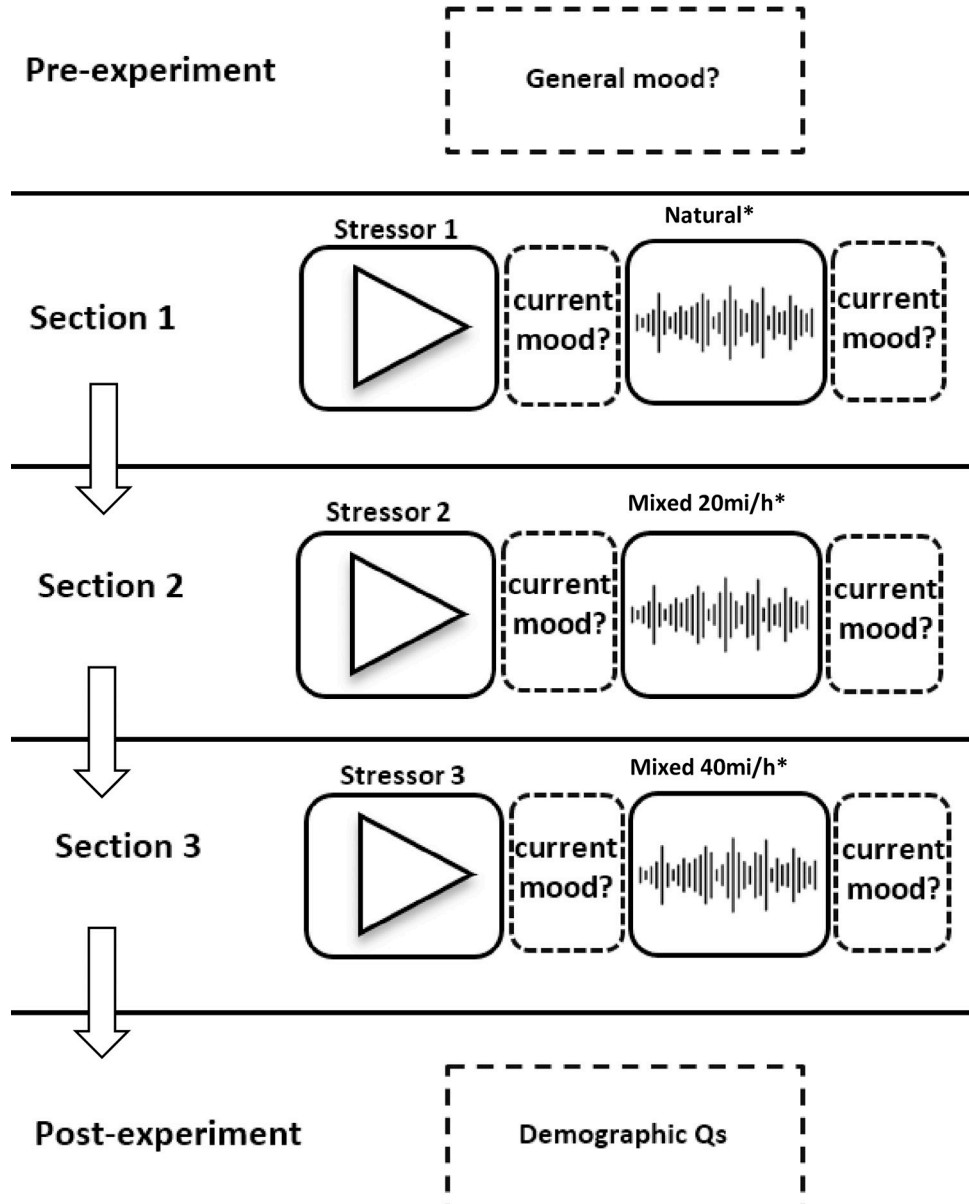

**Fig 1. Schematic of experimental design, including pre-experimental playback period, where 'trait' measures of general mood were recorded for each participant, sections 1, 2 and 3 of the experiment (soundscape playback) and post-experimental period, where questions on demographics and other general questions were asked.** The experiment contained 3 sections, each with a stressor video and soundscape recording playback followed by questions measuring subjective current mood. Playbacks were either a natural soundscape ('Natural') or a mixed natural and anthropogenic soundscape, with either 20 mi/h ('Mixed 20') or 40 mi/h ('Mixed 40') traffic sounds added. Sections marked with a * were randomly rotated between the experiment sections for each participant, but stressor video order remained the same.

Participants were asked to use noise cancelling (over the ear) headphones if available, or ear buds (in the ear) or laptop/computer speakers if they did not have access to headphones. Before starting the experiment, participants were asked about their mood in general (see Subjective Measures below), and then asked to do a sound test. During the sound test, participants were asked to put on their headphones, and click a test link that played some sounds of human

talking. They were then asked to adjust their headphones to a level that was comfortable, but loud enough to immerse themselves in the sounds. Talking was chosen as the test sound as it was sufficiently different from the sounds used in the main experiment.

The experiment was comprised of three sections (Fig 1). In each section, each participant was exposed to a stressor video and then asked to rate their current mood (see Subjective Measures below). Participants were then exposed to one of the three soundscapes and again asked to rate their mood, allowing a measure of mood recovery. All participants were played all three stressor videos and all three soundscapes. The order of soundscapes was randomised between participants, but the stressor videos were not (Fig 1). Soundscapes were randomised to control for any order effects, with each participant exposed to all three soundscapes as part of the repeated measures design.

## Stressors

Stressors are often used in psychological research as a way of standardising mood state, before subjecting a participant to a condition [11,15], which in this case was a soundscape playback. We created three multi-modal cognitive stressors including arithmetic, based on methods of similar studies [11,15]. All stressors were 1-minute long video comprised of a written maths question, which was obscured by flashing between different colours of text and background, along with stressful sounds (e.g. either a squeaky noise, annoying music or an alarm beeping). Each video contained a different maths question, sound, and colour pallet. Stressor order was the same for each participant, due to constraints within the survey program design architecture, but we included stressor as a fixed effect in statistical analysis to control for any effect on subjective measures (see Statistical Analysis below).

## Subjective measures

In order assess participants mood before and during the experiment, we used two measures commonly used in psychological research and clinical settings to diagnose anxiety and measure acute shifts in mood. Two measures were used to ensure internal consistency in the experiment (i.e. if scores from the two measures mirrored each other, then they were both more likely a true representation of mood). We measured a participant's general mood as subjective 'trait' anxiety (STAI-T) before the experiment, using a validated short form of the State-Trait Anxiety Inventory (STAI) scale (S1 Fig) [28,29]. We also measured a participant's subjective mood changes (i.e. their current mood 'state') during the experiment, including after each of the three stressor videos and after each soundscape playback. We measured subjective mood as current 'state' anxiety, stress, and pleasure (hedonic tone). State anxiety is defined as a temporarily anxious emotional state influenced by the current situation, hedonic tone is a measure of pleasure in response to a situation, and subjective stress, the level of stress or tension currently felt. We measured current mood in terms of subjective stress and hedonic tone using a short form of the University of Wales Institute of Science and Technology Mood Adjective Checklist (UWIST MACL) [30]. Participants were asked to rate how they felt currently in terms of four mood items (relaxed, nervous, happy, sad) on a 4-point Likert scale (S2 Fig). Current stress was scored as nervous plus reverse relaxed scores and hedonic tone as happy plus reverse sad scores. Therefore, increased subjective stress and increased pleasure (hedonic tone) were represented by higher scores.

We also measured current mood in terms of anxiety (STAI-S) using a validated short form of the STAI scale (S3 Fig) [28,29]. Both STAI-S and STAI-T scales included three anxiety present and three anxiety absent items, rated on a 4-point Likert scale (S1 and S3 Figs). Both STAI-S and STAI-T scores were calculated as the sum of anxiety present items and reverse

anxiety absent items scores, so that higher scores represented increased current and general anxiety.

## Demographics and other participant information

Self-selected student participants are often used in studies of this type [15,31,32] However, despite students being generally regarded as appropriate subjects for this type of study [33], we wanted to understand any inherent biases that existed in the participant pool. We therefore collected a range of participant information before the main playback experiment to gain an understanding of the biases that may exists in our sample. We then chose a subset of that participant information for use in analysis (see *Statistical Analysis*), chosen as they were the most likely to influence a person to being more or less sensitive to natural soundscapes, which could in turn bias the results. We collected demographic data (age, ethnicity, gender), all commonly measured in similar studies examining wellbeing in response to soundscape playbacks (e.g. [10,13]). We also included several questions related to participants' relationship with their environment. We asked whether participants live, work and grew up in urban, semi-rural or rural environments and whether they had a preference for natural or urban environments (scale: none, slight, strong preference for either urban/rural), whether participants noticed sounds in their environment (4-point scale: not sure, sometimes, often, very often). We also included the type of listening device used to take part in the experiment (in ear or over ear headphones or speakers), as this might influence the ability of the participant to fully engage in the experiment. As bird sound was prominent in all three soundscapes, to rule out any effect of bird phobia, we also included a question asking about phobias in general and included birds in a list of other common phobias (e.g. bats, spiders).

## Statistical analysis

We analysed all data using the R package lme4 (v.1.1–26 [34]) in R Studio (R version 4.0.5) using generalized linear mixed models (GLMMs). We analysed three subjective mood measures datasets, collected as part of the main soundscape playback experiment. Subjective mood measures included *stress* and *hedonic tone* (UWIST MACL) and *anxiety* (STAI-S). We included participant information data included in the analysis a general (trait) anxiety measure (*STAI-T*), demographic variables (*age*, *gender*), inherent preferences for nature scores (*preference for natural environments*).

## Analysis of subjective mood measures

Response variables included *stress*, *hedonic tone* and *anxiety* scores calculated from scores recorded after each soundscape. Fixed effects included in the full model were soundscape (factor variable with levels: 'natural', 'mixed 20', 'mixed 40'), STAI-T (continuous variable), age (continuous variables), gender (factor coded as binary: 1 = male, 0 = female, NA = non-binary), preference for natural environments (factor coded as a binary score: 1 = slight and strong preference for natural environments, 0 = slight and strong preference for urban environments or no preference). Each participant was exposed to all stressors and all soundscapes. However, soundscape order was randomised within the Qualtrics software, whereas stressor order was not. We therefore included stressor identifier (factor variable with levels: A, B, C) as a fixed effect in the full model to control for any effect of stressor type or order on the outcome measures. We also chose to use mixed effect models due to their ability to include random effects (as well as fixed effects) and included the random effect of participant number. Using this mixed modelling method allowed us to control for order of the fixed factor effects

soundscape and stressor as well as any inherent variation in stress, hedonic tone and anxiety amongst participants.

## Modelling procedure

Final models were selected using a backwards step-wise model selection procedure to find the most parsimonious yet best fitting model. Fixed effect variable terms were removed sequentially, and likelihood ratio tests (LRTs) (ANOVA) were performed between models with and without that term. Variables that were significant in LRTs were retained in the final model and non-significant terms were removed. Models were also compared based on their second order Akaike Information Criterion (AICc) and the model with the lowest AICc that contained all significant terms was chosen as the final model. Relevant interaction effects were also tested in the same way. Tukey contrast tests were also performed on final models to test for differences between soundscape levels. Statistics are presented in tables, including model summary statistics (estimates and s.d. for fixed effects and variance and s.d for random effects), likelihood ratio tests ($\chi^2$, d.f. and $p$ value) and Tukey contrast statistics (estimate, SE, z and $p$ values). We tested whether residuals were normal for LMMs and validated all models using a simulation method in the Dharma package in R [35], which tested for homogeneity of variance, zero inflation and overdispersion (v0.4.5).

## Results

### Participant information and demographics

Participants (n = 68) were mainly white British female undergraduate in their first- or second year studying science and psychology/social science students, living and working in an urban environment (S1 Table). Participants had an age range of 18–42. Most participants (66.18%) either had a slight or strong preference for natural environments and 22.06% and 38.24% reported noticing sounds *sometimes* and *very often* in their environment. Most participants grew up in semi-rural (39.71%) or urban (44.12%) environments. Only 1.4% of participants had a bird phobia. Most participants used headphones (in ear 41.18%, over ear 36.76%) to listen to the soundscapes and 22% listened on laptop speakers.

### Subjective measures

Current stress and anxiety were lower and pleasure (hedonic tone) scores higher in participants after experiencing all three soundscapes when compared to the three stressors (Table 1, Fig 2). There was a significant effect of soundscape treatment on all three subjective measures (Table 2, see S3 Table for model selection statistics). Current stress and anxiety scores increased across soundscape treatments (Fig 2; S2 Table). We recorded significant differences between the 'natural' and 'mixed 40' treatments for current stress (UWIST MACL) and 'natural' vs 'mixed 20' and 'natural' vs 'mixed 40' treatments for current anxiety (STAI-S) (Table 3). Pleasure scores (hedonic tone) were lower after exposure to the 'mixed 40' when compared to both 'natural' and 'mixed 20', however this was marginally non-significant when analysed with post-hoc Tukey contrasts (Table 3), despite soundscape treatment being a significant fixed effect when analysed with a GLMM (S2 Table).

There was a significant positive trend between general 'trait' anxiety (STAI-T) and current stress (UWIST MACL) and anxiety (STAI-S) scores (Fig 3). However, pleasure (hedonic tone) decreased with increased general 'trait' anxiety (STAI-T) scores (Fig 3). General 'trait' anxiety (STAI-T) score was also significant when included as a fixed effect in GLMMs for all three measures (Tables 2 and S2).

**Table 1. Descriptive statistics (mean and s.d.) for subjective measures including UWIST MACL stress and hedonic tone (hedtone) and STAI state anxiety scores for 3 stressors and 3 soundscape treatments, including a natural soundscape and mixed natural and urban soundscapes with 20 and 40mi/h traffic noise ('natural', 'mixed 20', 'mixed 40').**

| Stressor/soundscape | Stress | | Hedtone | | Anxiety | |
|---|---|---|---|---|---|---|
| | Mean | s.d. | Mean | s.d. | Mean | s.d. |
| Stressor A | 5.87 | 1.68 | 5.16 | 1.30 | 15.66 | 3.87 |
| Stressor B | 5.12 | 1.44 | 5.35 | 1.29 | 14.94 | 4.00 |
| Stressor C | 4.82 | 1.41 | 5.62 | 1.39 | 14.63 | 3.93 |
| Natural | 3.03 | 1.06 | 6.26 | 1.46 | 9.57 | 2.86 |
| Mixed 20 | 3.28 | 1.05 | 6.32 | 1.10 | 10.87 | 3.24 |
| Mixed 40 | 3.54 | 1.25 | 6.01 | 1.17 | 11.32 | 3.27 |

Higher UWIST MACL stress and hedtone scores represent higher levels of subjective stress and pleasure respectively, and higher STAI state scores represent increased anxiety.

Stressor had no effect on current stress (UWIST MACL) or anxiety STAI-S scores, but pleasure (hedonic tone) scores were significantly different between stressors and so this variable was controlled for and retained in the final model for hedonic tone (S2 Table).

## Discussion

This online study set out to assess the effect on participants mood of listening to a natural soundscape including birdsong, following exposure to a multi-modal cognitive stressor, and whether the addition of anthropogenic soundscapes including road traffic of different speeds also affected mood recovery. We show that listening to a natural soundscape (including bird song) can reduce self-reported current stress and anxiety levels, and that mood recovery (after a stressor) is lessened on addition of road traffic soundscapes. Positive mood (pleasure/ hedonic tone) was also enhanced by the natural soundscape, but this was limited by traffic sounds. Results indicated that the natural soundscape alone was associated with the lowest levels of self-reported current anxiety and stress levels which then increased on listening to the mixed soundscape with 20 mi/hr road traffic, with the highest levels reported after the mixed soundscape with 40mi/h road traffic. Higher levels of pleasure (hedonic tone), reflecting positive mood, were reported after listening to the natural soundscape, but reduced in comparison after listening to the mixed natural and 40mi/h traffic soundscape.

### Do natural soundscapes improve mood?

When participants were exposed to the natural soundscape they had the lowest levels of subjective stress and anxiety relative to soundscapes including anthropogenic noise. This supports previous findings highlighting the positive impact that natural soundscapes can have on stress recovery and mental fatigue [8,14,36,37]. Our result therefore highlights the importance of the retention of suitable sized urban greenspace that are accessible to the public and large enough to support wildlife populations beyond the reach of anthropogenic pollutants such as traffic noise. This supports [38] who call for the urban populace to experience more robust, healthy and even wilder forms of nature, away from human disturbances. Access to greenspace rapidly reduces as cities grow, which reduces opportunities for people to experience nature [39], it is therefore essential that future urban development and expansion occurs with the provision of greenspace included to maximise the health and wellbeing benefits from these spaces.

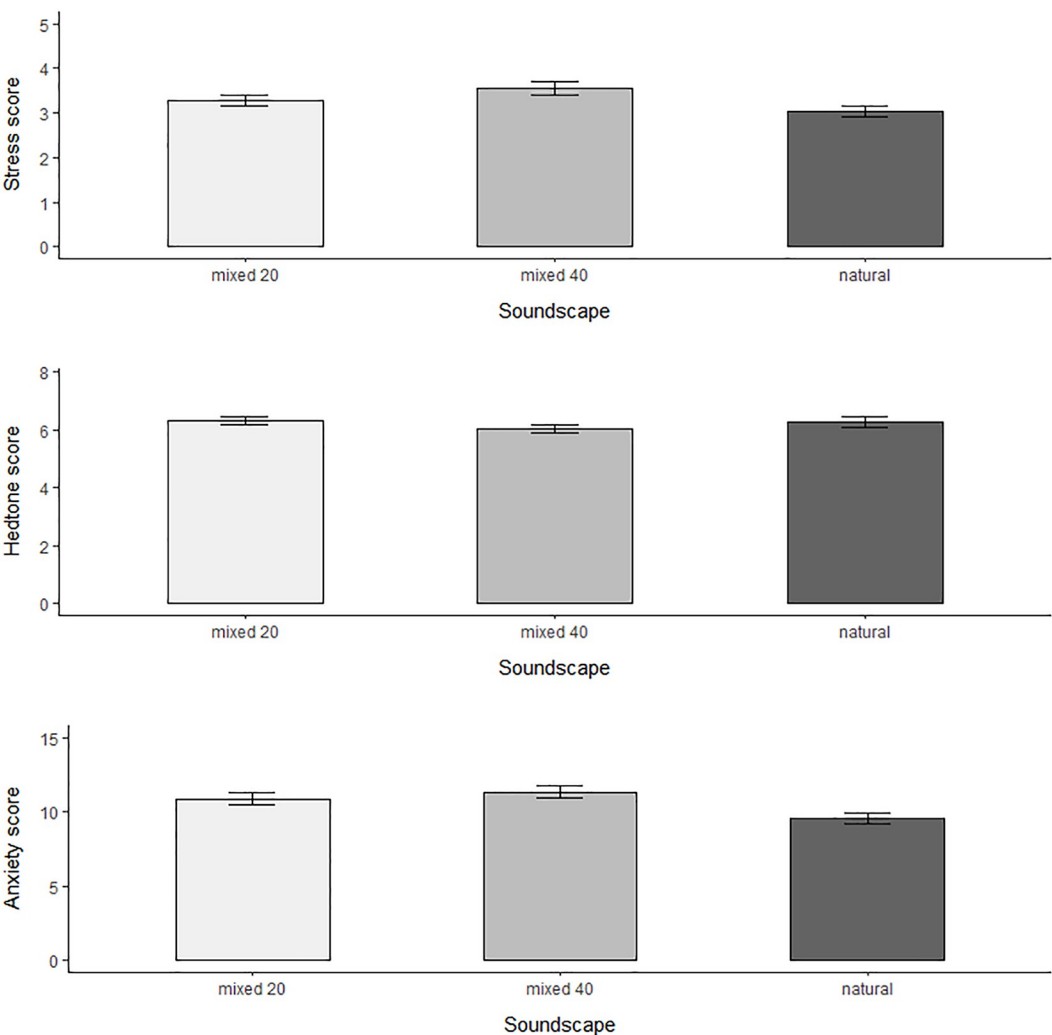

**Fig 2. Mean (±SE) for three subjective measure scores: UWIST MACL stress and hedonic tone (hedtone) and STAI state anxiety, for three soundscape treatments, including a natural soundscape (light grey bars) and mixed natural and urban soundscapes with 20 (medium grey bars) and 40mi/h (dark grey bars) traffic noise ('natural', 'mixed 20', 'mixed 40').** Stress (UWIST MACL) was scored as nervous plus reverse relaxed scores and hedonic tone as happy plus reverse sad scores. Therefore, increased subjective stress and increased (positive) hedonic tone were represented by higher scores. Anxiety scores were calculated from the STAI-S scale as the sum of anxiety present items and reverse anxiety absent items scores, so that higher scores represented increased anxiety. Significance stars are presented for Tukey contrasts (. = $p < 0.1$, * = $p < 0.05$, ** = $p < 0.01$, *** = $p < 0.001$).

## What are the benefits of reducing traffic speeds on mood?

Anthropogenic noise reduces the human ability to hear natural sounds, for example the ability of ornithologists to detect birds (e.g. [40]). Our results show that the presence of traffic noise does mask the positive impact of a natural soundscape on stress and anxiety in participants and that this was irrespective of age, gender or a pre-disposed preference for natural environments. We found a trend of decreasing stress and anxiety scores, with a decrease in traffic speed across the conditions, despite not all treatment comparisons being significant during multiple comparison tests. For example, there was no significant difference in levels of subjective stress between hearing a natural soundscape and natural soundscape alongside 20mi/h road noise, in contrast to 40mi/h traffic where both stress and anxiety were heightened.

**Table 2. Likelihood ratio test statistics for GLMMs for three subjective measures: UWIST MACL stress and hedonic tone (hedtone) and STAI-S (state) anxiety, including fixed effects soundscape treatment, STAI trait score (STAI-T), preference for natural environments, gender, age and stressor.**

| Model | Fixed effects | $\chi^2$ | df | $p$ | Significance |
|---|---|---|---|---|---|
| Stress (UWIST) | Soundscape | 11.20 | 2 | p < 0.01 | ** |
| | STAI-T | 4.03 | 1 | 0.04 | * |
| | Pref natural | 0.25 | 1 | 0.62 | NS |
| | Gender | 0.16 | 2 | 0.92 | NS |
| | Age | 0.11 | 1 | 0.74 | NS |
| | Stressor | 1.81 | 1 | 0.18 | NS |
| Hedtone (UWIST) | Soundscape | 6.81 | 2 | p < 0.05 | * |
| | STAI-T | 4.01 | 1 | p < 0.05 | * |
| | Stressor | 5.24 | 1 | p < 0.05 | * |
| | Pref natural | 0.43 | 1 | 0.51 | NS |
| | Gender | 1.46 | 2 | 0.48 | NS |
| | Age | 0.10 | 1 | 0.75 | NS |
| Anxiety (STAI-S) | Soundscape | 22.34 | 2 | p < 0.001 | *** |
| | STAI-T | 6.14 | 1 | p < 0.05 | * |
| | Pref natural | 0.25 | 1 | 0.62 | NS |
| | Gender | 1.32 | 2 | 0.52 | NS |
| | Age | 1.22 | 1 | 0.27 | NS |
| | Stressor | 2.44 | 1 | 0.12 | NS |

Higher UWIST MACL stress and hedtone scores represent higher levels of subjective stress and pleasure respectively, and higher STAI state scores represent increased anxiety. Significance stars include

* $p < 0.05$, ** $p < 0.01$, ** $p > 0.001$, NS = non significant).

Pleasure was also reduced during the higher traffic speed condition, but we recorded no difference between the natural soundscape and the mixed 20mi/hr traffic condition and only small differences recorded in the other treatment comparisons. The reason for these results is likely the differing sensitivity of the two subjective measures in their ability to pick up the differences in mood caused by the treatments, which is why we chose to include two measures (e.g. UWIST MACL has less mood items to score than the STAI-state measure, which may make it

**Table 3. Tukey contrast statistics for three subjective measures, including UWIST MACL stress and hedonic tone (pleasure) and STAI-S (state) anxiety, for three soundscape treatments, including a natural soundscape ('natural') and mixed natural and urban soundscapes with 20 and 40mi/h traffic noise ('mixed 20', 'mixed 40').**

| Model | Contrast | Estimate | SE | z | $p$ | significance |
|---|---|---|---|---|---|---|
| Stress | Natural vs mixed 20 | 0.25 | 0.15 | 1.65 | 0.23 | NS |
| | **Natural vs mixed 40** | **0.51** | **0.15** | **3.39** | **p < 0.01** | ** |
| | Mixed 20 vs mixed 40 | 0.26 | 0.15 | 1.74 | 0.19 | NS |
| Hedtone | Natural vs mixed 20 | 0.02 | 0.14 | 0.18 | 0.98 | NS |
| | **Natural vs mixed 40** | **-0.29** | **0.14** | **-2.10** | **0.09** | . |
| | **Mixed 20 vs mixed 40** | **-0.31** | **0.14** | **-2.29** | **0.06** | . |
| Anxiety | **Natural vs mixed 20** | **1.29** | **0.37** | **3.49** | **p < 0.01** | ** |
| | **Natural vs mixed 40** | **1.75** | **0.37** | **4.71** | **p < 0.001** | *** |
| | Mixed 20 vs mixed 40 | 0.00 | 0.46 | 0.37 | 0.44 | NS |

Significance stars include * $p < 0.05$, ** $p < 0.01$, ** $p > 0.001$ and NS = non-significant).

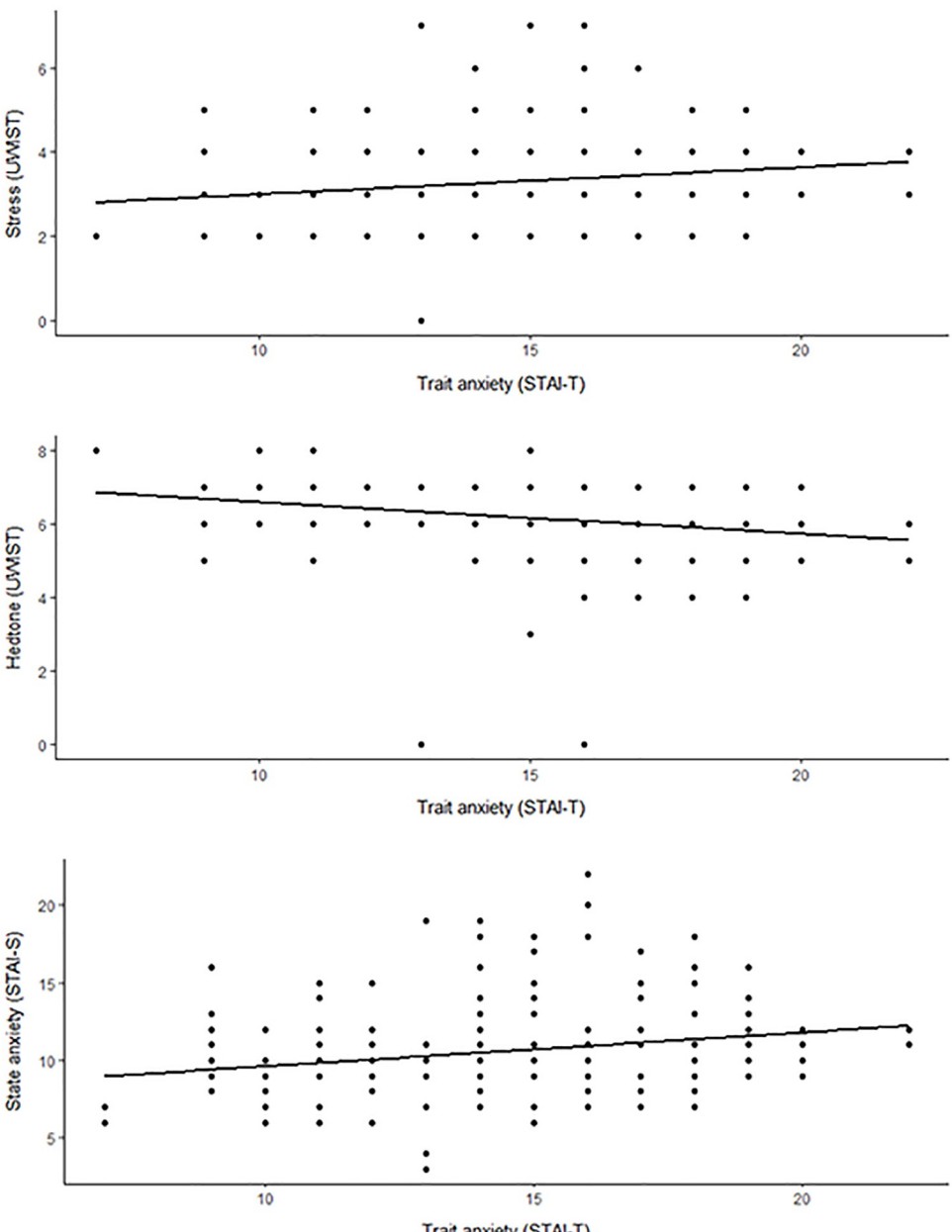

**Fig 3. Scatterplots of three subjective measures, including UWIST MACL stress and hedonic tone (hedtone) scores and STAI-S (state) anxiety score against STAI-T (trait) anxiety scores.**

comparatively more sensitive) (S1–S3 Figs). Despite some between treatment comparisons being non-significant, soundscape was a significant predictor of all three mood item scores and the trend was for higher traffic speeds to reduce mood recovery in comparison to when natural soundscapes were heard alone. Reduced traffic speeds can therefore be a positive intervention in improving accessibility to nature as well as reducing road injuries [41] and encouraging active play [26].

## How can reducing traffic speed influence health and wellbeing?

Anthropogenic noise pollution is associated with hypertension, cardiovascular disease risk [19,20], and an increased risk of mental health conditions, including depression and anxiety symptoms [21–23]. Our results show that for those with high baseline anxiety levels (so-called 'trait' anxiety STAI-T), exposure to anthropogenic soundscapes will lead to heightened levels of anxiety and stress, highlighting the pressures that many people will be exposed to on city streets on a daily basis. Trait anxiety (STAI-T) is a relatively stable characteristic reflecting the general background levels of anxiety experienced by individuals and is likely to be associated with other personality characteristics as well as mood states. Our results show that those participants with higher trait anxiety levels also reported higher current 'state' anxiety–how anxious they felt at the point of assessment, and similarly higher stress scores when compared to those with lower levels of trait anxiety who reported correspondingly lower state anxiety and stress. In addition, a significant proportion of people suffering from clinical levels of depression will also have anxiety [42]. A recent review of the mechanisms of anthropogenic noise impacts on mental health has pointed to a direct impact on the central nervous system, including adverse changes in brain tissue when people are exposed to traffic noise in their immediate environment every day [23]. Our study shows that exposure to natural soundscapes may alleviate some of the adverse effects on health and wellbeing caused by anthropogenic noise pollution. Therefore, the inclusion of wildlife and associated green environments within urban areas where over 80% of the UK population lives [43] may then benefit those suffering from these conditions, and possibly help prevent the transition of negative mood to a clinical level requiring treatment.

## Study limitations, research directions and recommendations

Like other studies of this kind, we used a student participant pool [15,31,32]. Although there is president for doing so among the literature [33], participants may have been predisposed to favour natural soundscapes, due to their age, sex or backgrounds. However, neither age, sex nor preference for the natural environment influenced mood response to the soundscapes, despite inherent biases existing in the participant pool. This study included playbacks including a number of different bird sounds making up a dawn chorus soundscape but did not change the level of biodiversity represented in the soundscape. Future studies could therefore include participants from a range of backgrounds and ages, with both sexes represented equally, to examine how participant socio-cultural background may influence responses to different elements of soundscapes, and whether prior knowledge of bird song influences response. Understanding how biodiversity impacts wellbeing via the sonic environment is an important avenue for future research, including the question of whether more biodiverse soundscapes are better at enhancing mood recovery. It is also important to understand the interaction between anthropogenic sounds and biodiversity in the sonic environment in relation to urban planning and effects on human health and wellbeing. From our study, it is clear that a reduction in urban speed limits would have benefits to human wellbeing in urban environments. Further research is also needed to understand how technological adaptations to the urban soundscape, such as the widespread transition to hybrid and electric vehicles, could reduce the impact of noise pollution on both human and wildlife in cityscapes. We recommend work that explores the redesign of major urban transit routes, both for potentially polluting motor vehicles and for pedestrian walkways or cycle paths, to include green spaces and significant vegetation that encourages wildlife. Greening in this way will in turn will dampen traffic noise, help absorb air born pollutants as well as benefitting mental health supporting a healthier travelling public [3,5,44].

## Conclusion

There is increasing pressure to implement reduced speed limits within cities, however speed reduction initiatives are still met with considerable resistance amongst policymakers and local communities. Here, we demonstrate that a reduction in traffic speed and therefore noise pollution can aid stress recovery and reduce anxiety highlighting the importance of exposing urban populations to wildlife. The reduced level of stress, anxiety and higher level of pleasure (hedonic tone) experienced by participants when exposed to a natural soundscape compared to a stressor event, even in the presence of masking anthropogenic sounds, highlight the importance of being able to hear natural sounds in our cities. City-wide strategies such as reducing road traffic speeds and conserving urban greenspace are therefore necessary steps to aid stress recovery and reduce anxiety.

## Supporting information

**S1 Doc. Survey example experienced by participants.**
(DOCX)

**S1 Fig. Example of scale presented to participants to rate their general mood (trait anxiety).**
(DOCX)

**S2 Fig. Example of scale presented to participants to rate their current mood in terms of stress and hedonic tone (pleasure).**
(DOCX)

**S3 Fig. Example of 4-point Likert scale presented to participants after each stressor video and soundscape file.**
(DOCX)

**S1 Table. Demographic data for all participants.**
(DOCX)

**S2 Table. GLMM statistics for subjective measures UWIST MACL stress and hedonic tone (pleasure) and STAI-S current state anxiety scores.**
(DOCX)

**S3 Table. Model selection statistics for GLMM for three subjective measures, UWIST MACL stress and hedtone and STAI state anxiety.**
(DOCX)

**S1 Data. Current state anxiety (STAI-S) dataset.**
(XLSX)

**S2 Data. UWIST MACL dataset (stress and hedonic tone).**
(XLSX)

## Acknowledgments

We would like to thank Gary Moore for the use of his bird song soundscape and for recording traffic noise for us.

## Author Contributions

**Conceptualization:** Lia R. V. Gilmour, Paul R. Lintott.

**Formal analysis:** Lia R. V. Gilmour.

**Funding acquisition:** Paul R. Lintott.

**Investigation:** Lia R. V. Gilmour.

**Methodology:** Lia R. V. Gilmour, Isabelle Bray, Chris Alford.

**Supervision:** Isabelle Bray, Paul R. Lintott.

**Writing – original draft:** Lia R. V. Gilmour.

**Writing – review & editing:** Isabelle Bray, Chris Alford, Paul R. Lintott.

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
