## [Decision Letter · Decision Letter 0]

24 May 2024

PONE-D-24-08824Natural Soundscapes Enhance Mood Recovery Amid Anthropogenic Noise PollutionPLOS ONE

Dear Dr. Lintott,

Thank you for submitting your manuscript to PLOS ONE. After careful consideration, we feel that it has merit but does not fully meet PLOS ONE’s publication criteria as it currently stands. Therefore, we invite you to submit a revised version of the manuscript that addresses the points raised during the review process.

We look forward to receiving your revised manuscript.

Kind regards,

Yuan Zhang, PhD

Academic Editor

PLOS ONE

Journal Requirements:

Reviewers' comments:

Reviewer's Responses to Questions

**Comments to the Author**

1. Is the manuscript technically sound, and do the data support the conclusions?

Reviewer #1: Partly

Reviewer #2: Yes

Reviewer #3: Partly

2. Has the statistical analysis been performed appropriately and rigorously? 

Reviewer #1: Yes

Reviewer #2: Yes

Reviewer #3: No

3. Have the authors made all data underlying the findings in their manuscript fully available?

Reviewer #1: Yes

Reviewer #2: Yes

Reviewer #3: Yes

4. Is the manuscript presented in an intelligible fashion and written in standard English?

Reviewer #1: Yes

Reviewer #2: Yes

Reviewer #3: No

5. Review Comments to the Author

Reviewer #1: It is an interesting paper that creatively included the investigations of the effect of traffic speeds on psychological responses to natural sounds. The study objectives are explicit, and the following works align well with such goals. The expression of this manuscript is logically clear and well-written, making the readers understand easily. However, I personally think the results of this study have not been exhibited, analyzed, and explained thoroughly. Also, the current contents are more narrative-like, lacking deep discussions of the phenomena found in the results. These aspects are really suggested for further improvement. Please see the detailed comments as follows.

Line 73-74

Due to the change in the subject of the description, the logical relationship before and after this sentence does not seem obvious. The link could be strengthened by modifying the formulation.

Line 81

Is it necessary to use the term “natural sounds” if the study only included and examined birdsongs?

Line 114

More details of the experiment are suggested to indicate clearly. For instance, how did the authors control the device condition, like sound volumes, of the participants’ sides? Is it possible that the conditions are different from person to person? Because it seems to be an online experiment conducted in this study, and the objects did not have to be somewhere (e.g., a room or lab) to participate in the experiment in person.

Line 171

The authors should indicate in this section why they preferred to use these models for data analysis.

Line 231

Some phenomena found in the results can be discussed more instead of simply describing them in this section. For instance, in Line 260, what is the possible reason for the lack of significant difference in stress levels between birdsong and birdsong with 20mi/h noise hearing?

Furthermore, the authors included some demographic factors in the analysis, but why were such factors not mentioned in this section? Even though they were insignificant in the models, the potential reasons or mechanisms can still be analyzed or speculated reasonably.

Line 235

The wording should be more concrete here. The present study does indicate the effect of birdsongs, a part/type of natural sounds/soundscapes, on stress and anxiety. However, the other types of natural sounds or soundscapes were not examined in this work, so this general term is not suggested here. Please check the whole manuscript to see whether there are similar cases.

Line282

The limitations of this study should also be stated.

Reviewer #2: The manuscript presents an interesting study assessing to what extend self-reported anxiety, stress and pleasure are affected by natural soundscapes (bird song) and traffic noise at different speed limits. The paper is well written and might be a relevant contribution to the field because the effect of speed limit has been largely understudied. The provided evidence is also very useful for urban planning. Overall, this paper is of high quality. There are some minor points to be considered before its publication:

Introduction is well-written and concise. However, the first paragraph is very broad. I would suggest to reduce or even omit it, and expand on the actual focus of the study, so to broaden on the soundscape part (paragraph two) on how soundscapes have been used in health and well-being studies before, while strengthening on what contributions and novelty the presented approach brings.

Lines 83-87: You also test for the effect of different background characteristics such as nature preference, pleas add here. I think adding a short paragraph on how socio-cultural characteristics have been studied before in soundscape-well-being studies will also justify more clearly the chose of variables in the questionnaire

Line 114: In the experimental design section, please state if randomization was used and in what ways. This information can be found only in the Supplementary, but I think it is important to clarify in the methods main text as well. In fact, figure S1 of the experimental procedure is really informative and clear and I would suggest including it in the main body of the article, not as part of the Supplementary, if possible

Stressors: if this is an established and validated stressor method, please add the reference

Line 165, Why did you ask about where participants grew up, please justify and add reference? In general, related to my previous comment, it would be better to connect the questions you have used with previous literature to justify the choice of your socio-demographic variables. Please add references and relate previous research studying such associations wherever possible

In the statistical analysis, did you somehow account for the order of the sound interventions (bird, bird + 20mi/h, bird + 40mi/h)? although they were randomized, accounting for the order effect in the model is important

Line 182: What do you mean by “non-experimental data” isn’t all data used in the experiment? Also, it is unclear in the methods description why you chose some of the demographic characteristics to be included in the models and others are omitted. In general, the difference in the aim/research question/hypothesis between the LMM and GLMM model is not very clear, especially the GLMM needs better explanation.

Line 184 -185 Why do you treat listening device (especially the latter) as a response variable? Please clarify

The results and discussion are well-written and concise. However, please add a short paragraph discussing study limitations. Please also add future directions for research, currently future recommendations are related only to planning

Reviewer #3: The study examines the impact of natural soundscapes, such as bird song, and the addition of traffic noise at different speeds (20 mi/h and 40 mi/h) on mood in urbanized environments where green spaces are scarce and exposure to anthropogenic noise is high.

Reviewer Comments:

1.The current review lacks sufficient depth regarding conclusions drawn from previous studies that combine traffic noise with natural sounds. Please expand the review to include a more comprehensive analysis of existing literature on this topic.

2.The manuscript does not provide a clear rationale for selecting 68 participants. Please elaborate on the selection criteria and the methods used to determine and verify the adequacy of this sample size.

3.There is a lack of detail regarding the duration of rest periods for subjects under different conditions. Please specify how long the subjects rested in each condition.

4.It would be beneficial to include Figure S1 in the main text and to add photos of the experimental scenes to provide clearer context and enhance the reader's understanding.、

5.The manuscript should clarify the relationships between the various scales used. Specifically, provide details on the relationship between STAI-T and STAI, and how these scales are interrelated in the context of the study.

6.The horizontal axis of Figure 2 is not clearly labeled. Please provide a clear and descriptive label for the horizontal axis.

7.The study mentions audio collection on two speed-limited roads but does not detail the traffic volume and vehicle type ratio on these roads. Please include this information and discuss whether it was considered during audio collection.

6. PLOS authors have the option to publish the peer review history of their article (what does this mean?). If published, this will include your full peer review and any attached files.

Reviewer #1: No

Reviewer #2: No

Reviewer #3: No

---

## [Author Response · Author response to Decision Letter 0]

12 Aug 2024

Response to Reviewers' comments

We have provided a response in bold underneath each reviewer’s comment and indicated the line number of any changes in the revised manuscript. Excepts are provided for ease of reference to changes in the manuscript. 

Reviewer #1 (R1)

It is an interesting paper that creatively included the investigations of the effect of traffic speeds on psychological responses to natural sounds. The study objectives are explicit, and the following works align well with such goals. The expression of this manuscript is logically clear and well-written, making the readers understand easily. However, I personally think the results of this study have not been exhibited, analyzed, and explained thoroughly. Also, the current contents are more narrative-like, lacking deep discussions of the phenomena found in the results. These aspects are really suggested for further improvement. Please see the detailed comments as follows.

We thank R1 for their positive comments regarding the study objectives and we welcome their feedback on the way the results have been presented, analysed and discussed. We believe the changes made in line with the feedback from R1 has made the manuscript a stronger contribution to the field and more widely applicable.

Line 73-74

Due to the change in the subject of the description, the logical relationship before and after this sentence does not seem obvious. The link could be strengthened by modifying the formulation.

We agree, and have moved the line lower down the paragraph to L81-83

Line 81

Is it necessary to use the term “natural sounds” if the study only included and examined birdsongs?

We thank the reviewer for picking this up. We have reviewed the whole manuscript and made our wording consistent throughout. 

Our study included soundscapes created from a natural soundscape recorded at a rewilded site during dawn chorus, including bird song and an urban soundscape including traffic noise recorded on two roads, one with 20mi/h traffic and one with 40 mi/hr traffic. A soundscape is different from sounds e.g. bird song or car noises recorded in isolation, as it represents the multi-layered soundscape you get with birds singing at different distances and the echoes from objects in the environment, all contributing to the experience of the soundscape. 

However, we agree the use of the term “natural sounds” was misleading. We have removed the reference to natural sounds throughout the manuscript and made sure we refer to natural soundscapes throughout. We have also included ‘including bird song’, or ‘including 20/40 mi/hr traffic noise’ where appropriate, to make it clear that the natural soundscapes included bird song and the anthropogenic/urban soundscape included traffic noise. 

We have also changed the names of the treatments throughout the manuscript, for clarity. The conditions are now Natural soundscape (previously bird only) and mixed 20 and mixed 40 treatments for the mixed soundscapes with natural soundscapes with 20 and 40 mi/h traffic noise added (previously bird + 20 and bird + 40 mi/hr). 

Line 114

More details of the experiment are suggested to indicate clearly. For instance, how did the authors control the device condition, like sound volumes, of the participants’ sides? Is it possible that the conditions are different from person to person? Because it seems to be an online experiment conducted in this study, and the objects did not have to be somewhere (e.g., a room or lab) to participate in the experiment in person.

We thank R1 for raising this. We did indeed attempt to control the sound volume as much as possible given that participants were doing the experiment at home at their computers or laptops. We asked that participants use ear cancelling headphones if available, or in ear headphones at the least. We also did a sound test before starting the experiment. We have added these details into the methods. 

L138-144: “Before starting the experiment, participants were asked about their mood in general (see Subjective Measures below), and then asked to do a sound test. During the sound test, participants were asked to put on their headphones, and click a test link that played some sounds of human talking. They were then asked to adjust their headphones to a level that was comfortable, but loud enough to immerse themselves in the sounds. Talking was chosen as the test sound as it was sufficiently different from the sounds used in the main experiment.”

Line 171

The authors should indicate in this section why they preferred to use these models for data analysis.

We used mixed models in both cases as it allowed us to include random effects in our models. This allowed us to control for any inherent mood differences among participants via the random effect of participant number. For clarity we have included the following on L228-230: 

“Using this mixed modelling method allowed us to control for order of the fixed factor effects soundscape and stressor as well as any inherent variation in stress, hedonic tone and anxiety amongst participants.” 

Line 231

Some phenomena found in the results can be discussed more instead of simply describing them in this section. For instance, in Line 260, what is the possible reason for the lack of significant difference in stress levels between birdsong and birdsong with 20mi/h noise hearing?

We agree that more detail examining the results would be helpful in determining the wider narrative. We used two subjective mood state measures in this experiment, and state the reasoning for this in the methods L166-167:

“Two measures were used to ensure internal consistency in the experiment (i.e. if scores from the two measures mirrored each other, then they were both more likely a true representation of mood).”

All three measures of mood- subjective anxiety, stress and hedonic tone (pleasure) were significantly different between soundscape treatments. However, on closer comparison, STAI anxiety scores show a significant difference between treatments, e.g. natural soundscape and mixed 20mi/hr, whereas UWIST MACL stress scores do not. There may be some power loss due to accounting for multiple comparisons. More likely is a difference in sensitivity between the tests. We have included this explanation and included reference to the supplementary figures showing the mood items in each Likert scale. Both measures show the same trend however, but one is perhaps more sensitive in picking up the nuances of mood differences felt by participants between the treatments. We have added an explanation to L305-320. 

Furthermore, the authors included some demographic factors in the analysis, but why were such factors not mentioned in this section? Even though they were insignificant in the models, the potential reasons or mechanisms can still be analyzed or speculated reasonably.

We thank R1 for the suggestion and have added in reference to the demographic factors in the aims section of the introduction and into the discussion. We have also added a discussion of the limitations of the study and included reference to demographics and other participant information. 

Line 235

The wording should be more concrete here. The present study does indicate the effect of birdsongs, a part/type of natural sounds/soundscapes, on stress and anxiety. However, the other types of natural sounds or soundscapes were not examined in this work, so this general term is not suggested here. Please check the whole manuscript to see whether there are similar cases.

We thank R1 for their comment and refer them back to our earlier response regarding use of the term natural soundscapes vs natural sounds. We acknowledge that we did not test other types of natural soundscapes, and have made this clear throughout the manuscript. 

Line282

The limitations of this study should also be stated.

We have added a section at the end of the discussion that includes limitations of the current study, future research directions and recommendations based on this study. L343-366

Reviewer #2 (R2)

The manuscript presents an interesting study assessing to what extend self-reported anxiety, stress and pleasure are affected by natural soundscapes (bird song) and traffic noise at different speed limits. The paper is well written and might be a relevant contribution to the field because the effect of speed limit has been largely understudied. The provided evidence is also very useful for urban planning. Overall, this paper is of high quality. There are some minor points to be considered before its publication:

We thank R2 for their positive comments about the manuscript, especially that it is of high quality and will have use in urban planning. 

Introduction is well-written and concise. However, the first paragraph is very broad. I would suggest to reduce or even omit it, and expand on the actual focus of the study, so to broaden on the soundscape part (paragraph two) on how soundscapes have been used in health and well-being studies before, while strengthening on what contributions and novelty the presented approach brings.

We thank R2 for this suggestion and have reduce paragraph one in the introduction accordingly. We have also expanded on paragraph two, including examples of studies including exposure to natural sounds and anthropogenic sounds. We have also made it clearer how our study is novel in comparison to what has been done before. 

Lines 83-87: You also test for the effect of different background characteristics such as nature preference, pleas add here. 

I think adding a short paragraph on how socio-cultural characteristics have been studied before in soundscape-well-being studies will also justify more clearly the chose of variables in the questionnaire

We have added the line: 

“We also tested whether general mood state, age, gender and inherent preference for natural environments affected mood recovery in response to natural and mixed natural and anthropogenic soundscapes.” L92-94

We have also added in a line introducing the influence of demographic and other factors in determining response to nature and natural soundscapes:

“Responses to nature, including natural soundscapes, may also be dependent on factors such as age, sex and socio-cultural experience e.g. childhood connection to nature (Hughes et al. 2018, Shu and Ma, 2020, Ge et al. 2023), though more research is needed to understand and tease apart the influence of these factors.” L53-56 

Line 114: In the experimental design section, please state if randomization was used and in what ways. This information can be found only in the Supplementary, but I think it is important to clarify in the methods main text as well. In fact, figure S1 of the experimental procedure is really informative and clear and I would suggest including it in the main body of the article, not as part of the Supplementary, if possible

We agree and have included Figure S1 as a figure in the main text (now Figure 1). We have also included the following line for clarity on randomisation:

“The order of soundscapes was randomised between participants, but the stressor videos were not (Figure 1). Soundscapes were randomised to control for any order effects, with each participant exposed to all three soundscapes as part of the repeated measures design.” L149-151

Stressors: if this is an established and validated stressor method, please add the reference

We based our stressor method on similar previous studies. Reference to these studies has been added to the methods section. L154

Line 165, Why did you ask about where participants grew up, please justify and add reference? In general, related to my previous comment, it would be better to connect the questions you have used with previous literature to justify the choice of your socio-demographic variables. Please add references and relate previous research studying such associations wherever possible

We included this question to understand the bias that may exist in the participant pool we were sampling from, i.e. students that were at university in an urban environment. Many similar studies in fields related to this research use student participants, however, we wanted to be able to discuss the limitations of our participant sample. We have now added a section discussing limitations and referenced these factors. We have also clarified the collection of participant information in the methods. 

We have added a section discussing limitations, future research and recommendations from the study. L343

In the statistical analysis, did you somehow account for the order of the sound interventions (bird, bird + 20mi/h, bird + 40mi/h)? although they were randomized, accounting for the order effect in the model is important

The order of soundscapes was randomised between participants, but the stressor videos were not (Figure 1). Soundscapes were randomised to control for any order effects, with each participant exposed to all three soundscapes as part of the repeated measures design.” L149-151

“Each participant was exposed to all stressors and all soundscapes. However, soundscape order was randomised within the Qualtrics software, whereas stressor order was not. We therefore included stressor identifier (factor variable with levels: A, B, C) as a fixed effect in the full model to control for any effect of stressor type or order on the outcome measures. We also chose to use mixed effect models due to their ability to include random effects (as well as fixed effects) and included the random effect of participant number. Using this mixed modelling method allowed us to control for order of the fixed factor effects soundscape and stressor as well as any inherent variation in stress, hedonic tone and anxiety amongst participants.” L222-230 

Line 182: What do you mean by “non-experimental data” isn’t all data used in the experiment? 

By this we mean any data collected before the experimental playback. We have made this clearer throughout the methods and results sections. 

Also, it is unclear in the methods description why you chose some of the demographic characteristics to be included in the models and others are omitted. 

In general, the difference in the aim/research question/hypothesis between the LMM and GLMM model is not very clear, especially the GLMM needs better explanation.

We have rewritten the methods for the statistical analysis to make the whole section clearer and included an explanation of why the chosen variables were used in the analysis. We have also added some info that was missing into the Participant info and demographics section in the results. 

Line 184 -185 Why do you treat listening device (especially the latter) as a response variable? Please clarify

We have removed this section as it was confusing. The main analysis includes a subset of “participant information” including demographic data, trait anxiety scores and preference for natural environments (see previous points). 

The results and discussion are well-written and concise. However, please add a short paragraph discussing study limitations. Please also add future directions for research, currently future recommendations are related only to planning

We have added a section discussing limitations, future research and recommendations from the study. L343

Reviewer #3 (R3)

The study examines the impact of natural soundscapes, such as bird song, and the addition of traffic noise at different speeds (20 mi/h and 40 mi/h) on mood in urbanized environments where green spaces are scarce and exposure to anthropogenic noise is high.

We thank R3 for their comments and recommendations to improve the manuscript. 

Reviewer Comments:

1.The current review lacks sufficient depth regarding conclusions drawn from previous studies that combine traffic noise with natural sounds. Please expand the review to include a more comprehensive analysis of existing literature on this topic.

We have expanded the introduction to include a more thorough examination of previous studies.

2.The manuscript does not provide a clear rationale for selecting 68 participants. Please elaborate on the selection criteria and the methods used to determine and verify the adequacy of this sample size.

Similar studies had between 40 and around 180 participants (see examples below), so w

---

## [Decision Letter · Decision Letter 1]

20 Sep 2024

Natural Soundscapes Enhance Mood Recovery Amid Anthropogenic Noise Pollution

PONE-D-24-08824R1

Dear Dr. Lintott,

We’re pleased to inform you that your manuscript has been judged scientifically suitable for publication and will be formally accepted for publication once it meets all outstanding technical requirements.

Kind regards,

Yuan Zhang, PhD

Academic Editor

PLOS ONE

Additional Editor Comments (optional):

Reviewers' comments:

Reviewer's Responses to Questions

**Comments to the Author**

1. If the authors have adequately addressed your comments raised in a previous round of review and you feel that this manuscript is now acceptable for publication, you may indicate that here to bypass the “Comments to the Author” section, enter your conflict of interest statement in the “Confidential to Editor” section, and submit your "Accept" recommendation.

Reviewer #1: All comments have been addressed

Reviewer #2: All comments have been addressed

Reviewer #3: All comments have been addressed

2. Is the manuscript technically sound, and do the data support the conclusions?

Reviewer #1: Yes

Reviewer #2: Yes

Reviewer #3: Yes

3. Has the statistical analysis been performed appropriately and rigorously? 

Reviewer #1: Yes

Reviewer #2: Yes

Reviewer #3: Yes

4. Have the authors made all data underlying the findings in their manuscript fully available?

Reviewer #1: (No Response)

Reviewer #2: Yes

Reviewer #3: Yes

5. Is the manuscript presented in an intelligible fashion and written in standard English?

Reviewer #1: Yes

Reviewer #2: Yes

Reviewer #3: Yes

6. Review Comments to the Author

Reviewer #1: (No Response)

Reviewer #2: (No Response)

Reviewer #3: (No Response)

7. PLOS authors have the option to publish the peer review history of their article (what does this mean?). If published, this will include your full peer review and any attached files.

Reviewer #1: No

Reviewer #2: No

Reviewer #3: No

---

## [Editor Report · Acceptance letter]

25 Oct 2024

PONE-D-24-08824R1 

PLOS ONE

Dear Dr. Lintott, 

I'm pleased to inform you that your manuscript has been deemed suitable for publication in PLOS ONE. Congratulations! Your manuscript is now being handed over to our production team.

Kind regards, 

on behalf of

Professor Yuan Zhang 

Academic Editor

PLOS ONE